# Expanding Role of Interleukin-1 Family Cytokines in Acute Ischemic Stroke

**DOI:** 10.3390/ijms251910515

**Published:** 2024-09-29

**Authors:** Paulina Matys, Anna Mirończuk, Aleksandra Starosz, Kamil Grubczak, Jan Kochanowicz, Alina Kułakowska, Katarzyna Kapica-Topczewska

**Affiliations:** 1Department of Neurology, Medical University of Bialystok, Marii Skłodowskiej-Curie 24A, 15-276 Białystok, Poland; paulina.matys@sd.umb.edu.pl (P.M.);; 2Department of Regenerative Medicine and Immune Regulation, Medical University of Bialystok, Waszyngtona 13, 15-269 Bialystok, Poland

**Keywords:** stroke, interleukin, cytokines, inflammation, immunology

## Abstract

Ischemic stroke (IS) is a critical medical condition that results in significant neurological deficits and tissue damage, affecting millions worldwide. Currently, there is a significant lack of reliable tools for assessing and predicting IS outcomes. The inflammatory response following IS may exacerbate tissue injury or provide neuroprotection. This review sought to summarize current knowledge on the IL-1 family’s involvement in IS, which includes pro-inflammatory molecules, such as IL-1α, IL-1β, IL-18, and IL-36, as well as anti-inflammatory molecules, like IL-1Ra, IL-33, IL-36A, IL-37, and IL-38. The balance between these opposing inflammatory processes may serve as a biomarker for determining patient outcomes and recovery paths. Treatments targeting these cytokines or their receptors show promise, but more comprehensive research is essential to clarify their precise roles in IS development and progression.

## 1. Introduction

Ischemic stroke (IS), a condition characterized by the interruption of blood flow due to arterial obstruction by thrombi or emboli, leads to brain tissue damage [1,2]. As the most predominant form of stroke, accounting for approximately 85% of all cases, it ranks as the second leading cause of death worldwide and poses a substantial threat to the quality of life for millions [3,4].

At its core, cerebral ischemia results from an inadequate blood supply to brain tissue. This deprivation of oxygen and glucose triggers a cascade of detrimental biochemical events. The disruption of cellular energetics and the ionic balance ultimately leads to oxidative stress, excitotoxicity, and cell death. The cumulative effect of these processes manifests as an area of irreversible brain damage, known as an infarct [5].

The affected tissue subsequently releases specific molecular signals, known as damage-associated molecular patterns (DAMPs), that set in motion a series of inflammatory cascades involving both innate and adaptive immune systems. Acting as biological alarms, DAMPs activate resident brain immune cells, particularly microglia. Once activated, glia cells adopt both pro-inflammatory and ani-inflammatory phenotypes. The integrity of the blood–brain barrier (BBB), initially compromised by the mechanical effects of the stroke itself, is further aggravated by an array of pro-inflammatory cytokines and chemokines, along with secreted matrix metalloproteinases (MMPs) and upregulated cell adhesion molecules. This disruption allows peripheral immune cells to infiltrate the brain tissue, which further amplify the inflammatory response. This creates a self-perpetuating cycle of neuronal death and immune activation that can extend the initial injury [5,6]; see Figure 1.

Interestingly, the post-ischemic inflammatory response exhibits a dual nature. While certain pro-inflammatory mediators can exacerbate tissue damage, other components, such as anti-inflammatory cytokines and growth factors, have neuroprotective functions and support recovery. Research findings indicate a link between the intensity of inflammatory reactions and the extent of cerebral damage [2,7]. Consequently, the delicate equilibrium between destructive and protective inflammatory processes is pivotal in determining the course of the post-ischemic recovery and patient’s long-term prognosis [8,9]. By analyzing the profile of inflammatory markers, particularly cytokines, clinicians may gain valuable insights into IS severity and outcomes. Furthermore, this understanding points to inflammation-related pathways as promising avenues for developing novel therapeutic interventions in IS management, potentially leading to more targeted and effective treatments.

The interleukin-1 (IL-1) family is a master regulator of inflammation, both in the acute and chronic state. IL-1 was the first interleukin to be discovered and the most studied one due to its strong proinflammatory effects. Further studies show that the IL-1 family is a diverse group of cytokines, comprising pro-inflammatory and anti-inflammatory members. This family continues to expand, with current knowledge comprising interleukin-1α (IL-1α), interleukin-1β (IL-1β), interleukin-18 (IL-18), interleukin-33 (IL-33), several interleukin-36 (IL-36) isoforms, interleukin-37 (IL-37), and interleukin-38 (IL-38), along with two inhibitors, interleukin-1 receptor antagonist (IL-1Ra) and interleukin-36 receptor antagonist (IL-36Ra) [10]; see Table 1. These cytokines have diverse cellular origins, as seen in Table 1, and far-reaching effects that address a wide range of health issues, where immune responses can initiate or exacerbate disease processes, including asthma, rheumatoid arthritis, atherosclerosis, inflammatory bowel disease, psoriasis, and type two diabetes [11].

In the aftermath of IS, the brain’s inflammatory response is rapidly activated, with IL-1 family members at the forefront of this cascade. These molecules orchestrate a delicate balance between promoting and suppressing inflammation, which is particularly relevant in the context of IS [8,11]; see Figure 2.

As the IL-1 family’s role in IS continues to evolve, it opens up new avenues for diagnostic and therapeutic strategies. This review synthesizes current knowledge on the IL-1 family’s role in IS modulation and identifies promising IL-1-related biomarkers. Additionally, it discusses future directions for IS management, emphasizing the IL-1 family’s potential in developing novel therapeutic interventions.

## 2. Interleukin-1 (IL-1)

### 2.1. IL-1α and IL-1β: Key Players in Ischemic Stroke-Induced Inflammation

IL-1, encompassing IL-1α and IL-1β, orchestrates a sophisticated immune response that significantly influences IS progression and outcomes [27].

While sharing similar biological effects, IL-1α and IL-1β exhibit distinct patterns of activation and expression in the context of IS. IL-1α is rapidly released from damaged cells, kickstarting the sterile inflammatory process predominantly near areas of focal neuronal damage and BBB [13,28,29,30]. Its expression in the brain precedes that of IL-1β, which highlights IL-1α‘s role as an early mediator of inflammation [31]. In contrast, IL-1β ramps up more gradually. Its expression increases within hours of IS onset and can persist for days, contributing to both acute and chronic inflammatory responses [14,27].

Both IL-1α and IL-1β are generated by activated microglia, but the supplementary sources of these cytokines vary. Accumulating platelets at the site of injury contribute to IL-1α production, while infiltrating leukocytes, migrating to the ischemic region in response to inflammatory signals, generate IL-1β [13,32].

### 2.2. IL-1α and IL-1β–Mechanisms of Action

IL-1α and IL-1β are pro-inflammatory cytokines that exert their biological effects primarily by binding to their shared receptor, IL-1 type 1 receptor (IL-1R1). This interaction prompts the recruitment of IL-1 receptor accessory protein (IL-1RAcP), also known as IL-1 type 3 receptor (IL-1R3), forming a receptor complex that triggers intracellular signaling cascades. The IL-1/IL-1R1/IL-1RAcP complex causes the dimerization of intracellular Toll-IL-1-receptor (TIR) domains, creating a docking site for myeloid differentiation primary response protein 88 (Myd88). Subsequently, additional signaling molecules are recruited, including IL-1R-associated kinases (IRAKs) and TNF-receptor-associated factor 6 (TRAF6). This engagement ultimately leads to the activation of the mitogen-activated protein kinase (MAPK) and nuclear factor kappa B (NF-κB) pathways [33,34]; see Figure 3.

The downstream effects of IL-1α and IL-1β signaling result in a multifaceted inflammatory response, encompassing the enhanced production of cytokines and chemokines, stimulation of matrix metalloproteinases (MMPs), and upregulation of adhesion molecule on endothelial cells [14]. 

The effects of IL-1 are further enhanced by the widespread distribution of IL-1R1 across various cell types, including those in peripheral tissues and in the central nervous system (CNS). Its expression on neurons, glial cells, and endothelial cells lining cerebral blood vessels is particularly important in the context of IS. This ubiquitous distribution enables a coordinated inflammatory response across multiple tissues and cell types, underscoring the systemic impact of IL-1 signaling [27,35].

### 2.3. Regulatory Mechanisms of IL-1 Signaling: The Roles of IL-1R2 and IL-1Ra

Nature has evolved regulatory mechanisms to control IL-1 signaling, with IL-1 type 2 receptor (IL-1R2), IL-1 receptor antagonist (IL-1Ra), and IL-38 playing key roles in this process. IL-1R2 functions as a non-signaling receptor due to its lack of a cytoplasmic TIR domain, sequestering IL-1α and IL-1β and preventing their binding to IL-1R1 [34]. 

Meanwhile, IL-1Ra competitively binds to IL-1R1, effectively blocking the interaction and subsequent signaling of IL-1α and IL-1β [27,36]. Additionally, IL-1Ra action is mimicked by IL-38 [37,38]. These multiple layers of regulation highlight the importance of tight control over IL-1 signaling in maintaining immune homeostasis; see Figure 3.

### 2.4. IL-1α’s Impact on Brain Damage and Recovery

Studies have shown that individuals experience elevated levels of IL-1α following IS. This increase is particularly pronounced in patients with IS with pre-existing conditions, such as atherosclerosis, diabetes, and infection, which can prime the inflammatory response. In these cases, higher concentrations of IL-1α are observed both systemically and locally within the brain’s ischemic areas. This amplified inflammatory response involves the increased expression of adhesion molecules on vessel walls, enhanced recruitment of immune cells and platelets, and greater production of inflammatory substances. Consequently, patients with elevated IL-1α levels experience more severe brain damage and poorer recovery outcomes [30,39,40,41,42].

Recent experimental research by Liberale et al. on rodent models has shed light on the potential benefits of inhibiting IL-1α in IS [43]. Selectively targeting and neutralizing IL-1α resulted in a marked decrease in activated macrophages in the ischemic core, indicating a dampened inflammatory response. Furthermore, the penumbra region, the salvageable tissue surrounding the infarct, exhibited decreased endothelial activation, as evidenced by the reduced expression of key endothelial adhesion molecules, such as P-selectin, Intercellular Adhesion Molecule 1 (ICAM-1), and Vascular Cell Adhesion Molecule 1 (VCAM-1). This downregulation of adhesion molecules limits the infiltration of inflammatory cells into the affected brain tissue. Consequently, mice treated with anti-IL-1α antibodies exhibited smaller infarct volumes compared to control groups, which was accompanied by improved neurological functions in the treated animals [43].

Building on these findings, further research conducted by Murata et al. has revealed that IL-1α plays a crucial role in modulating astrocyte reactivity and enhancing the expression of aquaporin-4 (AQP4), a protein channel involved in water transport [44]. The interaction among activated microglia-derived macrophage-like cells, astrocytic AQP4 expression, and IL-1α release has been identified as a crucial mechanism in the exacerbation of brain edema following IS. Animals with more severe symptoms exhibited significantly elevated levels of both AQP4 and IL-1α in the ischemic core compared to those with milder manifestations [44].

Moreover, a meta-analysis by Zou et al. has revealed intriguing associations between specific variations in the IL1A gene, particularly the IL-1alpha-889 polymorphism, and a heightened risk of IS [45]. 

### 2.5. IL-1 β’s Impact on Brain Damage and Recovery

IL-1β plays a significant role in the progression of brain damage following IS. Murray et al. reported that elevated IL-1β levels correlate with increased volumes of hypoperfused tissue and larger infarct areas [46]. Catană et al. revealed a strong positive correlation between IL-1β levels at hospital admission and IS severity, as measured by the National Institutes of Health Stroke Scale (NIHSS). Moreover, higher IL-1β concentrations were associated with increased patient mortality rates [47]. Interestingly, IL-1β levels vary among different IS subtypes. Two independent studies by Tuttolomondo et al. and Licata et al. found notably elevated levels of this cytokine in patients suffering from cardioembolic IS compared to other types [48,49]. Therefore, Rezk et al. suggested that IL-1β’s involvement in IS pathology may be more closely linked to the formation of blood clots, as it stimulates the expression of coagulation-related genes [50]. 

Research into IL-1β inhibition has revealed promising neuroprotective effects in IS models. The deletion of IL-1β and its receptor IL-1R1 in mice resulted in attenuated IS-induced brain injury [27]. 

It is suggested that genetic factors contribute to IL-1β’s role in IS pathology. A study focusing on the Polish population conducted by Gorący et al. has shown that individuals possessing the C allele of the IL-1β rs1143627 genetic variant may face a higher likelihood of experiencing IS, especially the large-vessel subtype [36]. However, the relationship among the IL-1β gene polymorphism, IL-1β production, and IS remains debated. Investigations by Ma et al. have reported that carriers of the T allele at position-511 of the IL-1β gene exhibit higher IL-1β levels [51]. According to Dziedzic et al. -TT homozygotes may be risk factor of IS due to small-vessel disease [52]. Rezk et al. additionally observed an association between different gene polymorphisms of IL-1 β at position-511 and the severity of neurological impairments [50]. In contrast, two independent studies by Yang et al. and Iacoviello et al., focusing on younger populations, have reported conflicting findings and revealed that the TT genotype of the IL-1β gene exhibits lower IL-1β levels and has a decreased risk of IS [53,54]. Nonetheless, studies by Zang et al. and Li et al. suggested no significant connection between IL-1β polymorphisms and IS risk [55,56]. The variability of these findings highlights the need for further research to elucidate the precise mechanisms by which genetic variations in the IL-1β gene influence IL-1β production and, consequently, IS risk. Understanding these genetic factors could potentially lead to more personalized approaches for IS prevention and treatment strategies.

### 2.6. IL-1Ra: Endogenous Regulator and Therapeutic Prospect in Ischemic Stroke

Numerous preclinical studies and meta-analyses have provided compelling evidence of the protective effects of IL-1Ra in IS, demonstrating that reduced levels of IL-1Ra result in exacerbated damage following ischemic events, leading to deteriorated neurological functions and enlarged infarct volumes [34,57]. Animals treated with IL-1Ra exhibit smaller infract sizes, diminished associated neurological deficits, superior functional outcomes, and enhanced survival rates compared to untreated subjects. Notably, the neuroprotective properties of IL-1Ra have been demonstrated even in the presence of comorbidities commonly associated with IS, such as obesity, hyperlipidemia, or insulin resistance [58]. 

Clinical trials evaluating the use of a recombinant human interleukin-1 receptor antagonist (rhIL-1Ra), known as anakinra, confirmed the improvements in patient outcomes [59]. On a molecular level, treatment with anakinra led to a decrease in inflammatory markers, including neutrophil and total white blood cell (WBC) counts, as well as a reduced concentration of C-reactive protein (CRP) and interleukin-6 (IL-6) [59,60,61]. Safety studies have shown that anakinra administration does not significantly alter levels of circulating immunoglobulins or complement components, indicating no increased susceptibility to infections [62]. 

While anakinra is indeed the most well-known and extensively studied IL-1-pathway inhibitor, it is important to note other agents targeting this pathway. Potential therapeutic options for mitigating the inflammatory cascade include bermekimab, an antagonist of IL-1α, and canakinumab, an antagonist of IL-1β. However, their roles and abilities to cross the BBB require further investigation [27].

## 3. Interleukin-18 (IL-18)

### 3.1. IL-18 Expression 

IL-18 is a pro-inflammatory cytokine produced by various cell types, including macrophages, microglia, peripheral blood mononuclear cells (PBMCs), and neurons; see Table 1 [17].

### 3.2. IL-18 Signaling Pathways

IL-18 initiates its pro-inflammatory effects by binding to the IL-18 receptor alpha chain (IL-18Rα) and subsequent recruitment of the IL-18 receptor beta chain (IL-18Rβ) [16,63]. The IL-18/IL-18Rα/IL-18Rβ complex then activates a signaling cascade that culminates in the activation of the MAPK and NF-κB pathways [16,17]. IL-18’s activity is regulated by a circulating molecule called IL-18-binding protein (IL-18BP), which sequesters free IL-18 and prevents its interaction with target cell receptors. This mechanism serves to reduce IL-18’s pro-inflammatory effects [16]. 

### 3.3. The Multifaceted Role of IL-18 in Immune Regulation

IL-18 is a key player in both innate and adaptive immune responses. Its primary function is stimulating interferon-gamma (IFN-γ) production in a wide range of immune cells, including CD4-positive (CD4^+^) T cells, CD8-positive (CD8^+^) T cells, macrophages, and natural killer (NK) cells. However, the pro-inflammatory effects of IL-18 are largely dependent on its synergistic interactions with other cytokines, particularly interleukin-12 (IL-12) or interleukin-15 (IL-15). These interleukins increase IL-18 receptor expression on cell surfaces, thereby enhancing cellular responsiveness to IL-18. This interaction is vital, as IL-18 alone not only fails to induce IFN-γ production, but also can have paradoxical effects. In the absence of IL-12 or IL-15, IL-18 may actually promote Type 2 helper T (Th2) cells differentiation and the secretion of the immunomodulatory cytokines interleukin-4 (IL-4) and interleukin-13 (IL-13) [16]; see Figure 4.

Beyond its role in cytokine production, IL-18 stimulates NK cells and promotes T cell differentiation, particularly favoring Type 1 helper T (Th1) responses. Additionally, IL-18 potentiates the killing capacity of NK cells and CD8^+^ T cells by improving their perforin- and FasL-dependent cytotoxicity mechanisms [16,17,63,64].

### 3.4. Elevated IL-18: Predictor of Ischemic Stroke Severity and Outcomes

IL-18, as a multifunctional pro-inflammatory cytokine, plays a significant role in atherosclerosis progression and accelerates inflammation within plaques. These cumulative effects contribute to atherosclerotic plaque instability, subsequently creating an environment conducive to IS occurrence [65]. A recent study by Martirosian et al. revealed that elevated serum levels of IL-18 and related inflammatory biomarkers are linked to a higher risk of IS [66]. According to a study and meta-analysis conducted by Hao et al., patients with IS have elevated levels of IL-18 compared to healthy controls, and these concentrations positively correlate with IS severity [17]. However, research examining the relationship between IL-18 levels and the infarct volume or functional outcome measured with the modified Rankin Score (mRS) have yielded conflicting results. Hao et al. have found no significant correlation [17], while Wang et al. have reported a positive association [67].

### 3.5. IL-18 in Myocardial Infarction: A Beacon for Ischemic Stroke Research

Similarly to IS, myocardial infarction (MI) arises from an obstruction in blood flow, resulting in ischemia and subsequent inflammatory responses. Notably, as presented by Wang et al., the administration of anti-IL-18 antibodies or mesenchymal stem cells overexpressing IL-18BP resulted in a reduction in the infarct size [68]. Given the parallels in pathophysiological mechanisms, it suggests that modulating IL-18 activity could offer therapeutic benefits for IS.

## 4. Interleukin-33 (IL-33)

### 4.1. Release of IL-33 in Ischemic Brain Injury

In response to ischemic brain injury, IL-33 is swiftly released from damaged CNS cells, particularly from a subset of oligodendrocytes. Therefore, this release acts as a nuclear alarmin, signaling tissue damage or stress and triggering immune responses in affected areas [20]. Moreover, the increase in IL-33 levels likely reflects the extent of brain damage and BBB disruption [69].

### 4.2. IL-33 Signaling Pathways

IL-33 exerts its effects by binding to its specific receptor ST2. A membrane-bound form of ST2 (ST2L) and IL-1RAcP initiate intracellular signaling cascade [70]. The soluble form of ST2 (sST2) acts as a decoy receptor by binding free IL-33, thereby preventing its interaction with the membrane-bound ST2L. This mechanism effectively inhibits the IL-33/ST2 signaling [71]. The IL-33/ST2 pathway can exert both pro-inflammatory and anti-inflammatory effects, with its ultimate impact on disease progression being context-dependent and not fully elucidated [70,72].

### 4.3. The Multifaceted Role of IL-33 in Ischemic Stroke

Current research indicates that in IS, IL-33 mitigates pro-inflammatory responses [73], by various mechanisms; see Figure 5. Luo et al. emphasized that IL-33 facilitates the transition of microglia to a state that supports anti-inflammatory effects [74]. Additionally, IL-33 effectively shifts the balance of T helper (Th) cells, favoring anti-inflammatory Th2 cells over pro-inflammatory Th1 cells, which was demonstrated by Korhonen et al. and Luo et al. [9,75]. Luo et al. also revealed that IL-33 suppress Th17 cell activity, which further contributes to its anti-inflammatory effects [9]. 

One of the key functions of IL-33 is its ability to modulate cytokine production. According to Yang et al., it stimulates both Th2 cells and microglia to produce anti-inflammatory cytokines, particularly interleukin-10 (IL-10) [76]. Furthermore, a study by Korhonen et al. revealed that IL-33 promotes the production of IL-4 and other cytokines that enhance the phagocytosis of damaged cells, which is vital for clearing cellular debris and promoting tissue repair in the aftermath of neurological insults [75]. The IL-33/ST2 pathway’s neurorestorating effects following IS are also linked to activated and mobilized lymphocytes, including regulatory T cells (Tregs) and group 2 innate lymphoid cells (ILC2s). Studies by Ito et al. and Guo et al. demonstrated that IL-33 enhances brain-resident Tregs, which express unique neural-related genes [77,78]. A recent study by Liu et al. revealed that IL-33 stimulates not only brain-resident Tregs, but also Treg populations in the spleen, suggesting both local and systemic immune response modulation [79]. These ST2-dependent Tregs exhibit neuroprotective properties by producing anti-inflammatory cytokines, like IL-10, interleukin-35 (IL-35), and transforming growth factor β (TGF-β) [79,80]. During cerebral inflammation, Tregs modulate astrogliosis and polarize microglia towards anti-inflammatory states through amphiregulin (AREG) production. AREG also activates neuronal epidermal growth factor receptors (EGFRs), contributing to improved outcomes in cerebral ischemia via the IL-33/ST2/Treg/AREG/EGFR signaling cascade, as presented by Guo et al. [78]. 

ILC2s, accumulating in the penumbra region, have garnered significant attention in recent neuroinflammation research by Zheng et al. The protective role of ILC2s, primarily through IL-4 production, results in a reduced infarction volume and the subsequent decreased severity of neurological symptoms [81].

A study by Liu et al. has also demonstrated that IL-33 administration under hypoxic conditions attenuates neuronal apoptosis by reducing levels of pro-apoptotic proteins, like cleaved caspase-3 and BAX, while simultaneously increasing expression of the anti-apoptotic protein Bcl-2 [79].

### 4.4. Protective Effects of IL-33/ST2 Signaling on Patients with Ischemic Stroke

The IL-33/ST2 signaling pathway aids in cell preservation and regeneration by reducing neuronal loss and supporting the brain’s innate repair mechanisms after injury or disease. Studies have shown that higher IL-33 levels correlate with better IS outcomes, including NIHSS scores below 6 and improved long-term functions [2]. Conversely, lower IL-33 levels are linked to an increased risk of IS recurrence [82].

Research on the relationship between IL-33 levels and the ischemic lesion size has produced mixed results. Liu et al. initially found that higher IL-33 levels correspond to larger ischemic lesions [83], while independent studies by Li et al. and Yang et al. reported an inverse relationship [2,76]. Interestingly, a later study by Li et al. detected no significant link between IL-33 levels and the lesion size [82].

IL-33’s role in hemorrhagic transformation (HT), a serious complication in IS, has also been studied. Patients with HT had higher serum IL-33 levels compared to healthy individuals. Nonetheless, lower IL-33 levels were associated with a greater likelihood of HT in the overall acute IS patient population, consistent with findings on the relationship between IS severity and IL-33 levels [69].

### 4.5. The Role of ST2 in Ischemic Stroke

ST2 plays a crucial role in the pathophysiology of IS [71]. Research by Yang et al. has shown that ST2L deficiency promotes a pro-inflammatory M1 microglial phenotype and suppresses the secretion of anti-inflammatory mediators, like IL-10. The inflammatory environment subsequently leads to an increased infarct volume [76].

The sST2, as a natural inhibitor of IL-33/ST2 signaling, has garnered attention as a potential biomarker of inflammatory conditions [71]. A clinical study conducted by Chen et al. has revealed that patients with IS exhibit markedly higher sST2 levels than healthy individuals. These elevated levels positively correlate with the infarct volume and IS severity, as assessed by NIHSS [84]. Furthermore, Wolcott et al. demonstrated that higher sST2 levels have been associated with increased mortality rates and a higher risk of HT in patients with IS [85]. This correlation extends to long-term outcomes as well, with higher sST2 levels linked to poorer prognoses at 90 days, as presented by Wolcott et al. and Krishnamoorthy et al. [85,86]. Contrary to the aforementioned results, an investigation led by Dieplinger et al. concluded that sST2 exhibited no significant association with functional endpoints [87]. This apparent discrepancy highlights the need for further research.

New applications for sST2 as biomarker have emerged from a recent study by Mechtouff et al., suggesting that sST2 levels may have predictive value for adverse clinical events in patients undergoing mechanical thrombectomy (MT) [88].

### 4.6. Therapeutic Potential

In the realm of potential therapeutic interventions, compounds that target the IL-33/ST2 pathway may hold promise. Jiang et al. demonstrated that celastrol promotes the polarization of microglia towards the anti-inflammatory M2 phenotype through the IL-33/ST2 pathway, potentially mitigating the damaging effects of inflammation in IS [89].

### 4.7. Genetic Insights

Genetic studies have further elucidated the role of IL-33 in IS, revealing protective polymorphisms within the IL-33 gene and its receptor pathway. Notably, Guo et al. identified the rs4742170 variant in the IL-33 gene as a protective factor against IS [90]. Meanwhile, several single nucleotide polymorphisms (SNPs) in the IL-33/ST2 axis, including rs10435816, rs7025417, rs11792633, and rs7044343, have been associated with a decreased risk of large-artery atherosclerotic IS in the study by Li et al. [91].

## 5. Interleukin-36 (IL-36)

IL-36 cytokines comprises three agonists: IL-36α, IL-36β, and IL-36γ. Recent research highlights the growing importance of these molecules in orchestrating both innate and adaptive immune responses [92,93]. They are expressed in various tissues and cell types, including keratinocytes, brain tissue, and immune cells, such as monocytes, macrophages, and dendritic cells [21]. IL-36 cytokines, well-established in psoriasis, have recently emerged as significant players in a spectrum of health conditions, spanning from acute to chronic manifestations. Ongoing research is continually expanding our understanding of their diverse roles [92,93,94,95].

### 5.1. IL-36 Signaling Pathways

These cytokines exert their effects by interacting with the IL-36 receptor (IL-36R), a heterodimer consisting of the IL-1 receptor-related protein 2 (IL-1Rrp2) and IL-1RAcP. Upon the binding of an IL-36 agonist to IL-36R, a series of molecular events is initiated, resulting in the activation of pro-inflammatory NF-κB and MAPK pathways [21]. The IL-36 signaling pathway is tightly regulated by two naturally occurring inhibitors: IL-36Ra and IL-38, competitively binding to the IL-1Rrp2 [21].

### 5.2. The Role of IL-36 in Myocardial Infarction and Ischemia-Reperfusion Injury

Despite recognizing the IL-36/IL-36R pathway’s significant pro-inflammatory role, our understanding of its biological functions in IS is still in its early stages.

However, recent research led by El-Awaisi et al. has shed light on its importance in ischemia–reperfusion injury. In the context of MI, IL-36 may exacerbate tissue damage through two mechanisms: promoting neutrophil recruitment, inducing inflammation in the microcirculation [92,93], and increasing the expression of adhesion molecules, such as ICAM-1 and VCAM-1 [96]. The use of IL-36Ra has also shown encouraging results, by mitigating neutrophil infiltration, improving blood flow dynamics, attenuating endothelial oxidative stress, and decreasing the expression of VCAM-1. Collectively, these effects contribute to a significant reduction in the size of infarcted areas [92,93].

The identification of the IL-36/IL-36R signaling pathway’s role in MI suggests its relevance in IS, giving a similar pathophysiological ischemic sequence.

## 6. Interleukin-37 (IL-37)

### 6.1. Expression Pattern of IL-37

IL-37 is expressed in various human cells and tissues, with primary sources being PBMCs and dendritic cells. Under normal physiological conditions, basal IL-37 levels are maintained at low concentrations. However, its expression can be significantly upregulated in response to various inflammatory stimuli and pro-inflammatory cytokines [23,97].

### 6.2. IL-37 Signaling Pathways

IL-37 initially engages with IL-18Rα. Unlike IL-18, IL-37 does not recruit the IL-18Rβ subunit. Instead, this initial binding triggers the association with the IL-1 receptor 8 (IL-1R8) also known as single immunoglobulin IL-1R-related receptor (SIGIRR) or Toll IL-1 receptor 8 (TIR8). The formation of this trimeric complex is essential for IL-37 to carry out its anti-inflammatory effects [16,23,97].

Interestingly, IL-18BP also has an affinity for IL-37. This interaction between IL-18BP and IL-37 potentially sequesters IL-37, thereby reducing its availability for anti-inflammatory signaling [16]; see Figure 6.

### 6.3. Role of IL-37 in Immune Regulation

IL-37 is known as a suppressor of both innate and adaptive immune responses, as well as inflammatory reactions triggered by various stimuli [97], thereby acting as a negative feedback mechanism to prevent excessive inflammation and tissue damage [23].

Specifically, IL-37 downregulates the production of pro-inflammatory cytokines, such as IL-1β, tumor necrosis factor α (TNF-α), and IL-6, by macrophages and dendritic cells. Furthermore, IL-37 inhibits the activation and maturation of antigen-presenting cells, thereby dampening their ability to initiate and propagate inflammatory cascades. IL-37’s influence on adaptive immune responses is shown by suppressing the activation of effector T cells while concurrently promoting the induction of Tregs [97].

### 6.4. IL-37 in Ischemic Stroke: A Paradoxical Player

Studies on IS have revealed complex and somewhat paradoxical findings regarding the cytokine IL-37. Despite its known anti-inflammatory properties, Zhang et al. revealed that elevated levels of IL-37 have been associated with less favorable prognoses. Patients with more severe neurological deficits, as indicated by an NIHSS score greater than 6, demonstrated higher circulating levels of IL-37 compared to those with moderate neurological impairment. A positive correlation was also observed between the infarct size and IL-37 levels [98]. Additionally, individuals diagnosed with large-artery occlusion exhibited significantly elevated circulating levels of IL-37 compared to individuals with small-artery occlusion [99]. Long-term follow-up studies provided further insights into the prognostic value of IL-37. Patients with higher IL-37 levels at the time of IS onset had poorer functional outcomes at 3 months, as assessed by a mRS score greater than 2. Moreover, elevated IL-37 levels were associated with an increased risk of IS recurrence within the third-month period [98].

Interestingly, higher circulating levels of IL-37 exhibited a positive correlation with elevated concentrations of IL-6 and CRP, which are typically indicative of heightened inflammatory states [98].

Genetically modified mice, expressing human IL-37, have provided valuable insights into IL-37’s protective role in IS. Dinarello et al. revealed reduced severity of inflammation in IL-37-transgenic (IL-37tg) mice [100]. In a study by Zhang et al., IL-37tg mice with IS exhibited less severe neurological deficits, smaller infarct volumes, and the reduced presence of pro-inflammatory macrophages compared to wild-type (WT) control mice. Additionally, the affected hemisphere displayed the elevated expression of anti-inflammatory markers, such as IL-10, IL-13, TGF-beta [101].

One plausible explanation for this counterintuitive observation is that the increased levels of IL-37 represent a compensatory response to the excessive inflammation induced by the ischemic events. Thus, the elevated IL-37 level is hypothesized by Liu et al. to indicate the degree of the inflammatory response rather than directly contributing to poor outcomes [102].

## 7. Interleukin-38 (IL-38)

IL-38 is recognized for its anti-inflammatory properties, acting as a counterbalance to inflammatory responses [103,104].

### 7.1. IL-38 Expression and Mechanisms of Action

This interleukin exhibits a widespread expression pattern across various organs, including the heart and brain. Notably, when cells undergo apoptosis, this process triggers the release of IL-38 into the surrounding environment [37]. IL-38 can competitively bind to IL-36R, interleukin-1 receptor accessory protein-like 1 (IL-1RAPL1), and IL-1R1. This binding ability is rooted in IL-38’s structural similarities to IL-1RA and IL-36RA [37,38]. As a result IL-38 mimics the action of both IL-1RA and IL-36RA, preventing the binding of the pro-inflammatory IL-1α, IL-1β, IL-36α, IL-36β, and IL-36γ to their respective receptors. Through this competitive inhibition, IL-38 effectively reduces the production of inflammatory mediators, thereby attenuating inflammatory responses [37,38,105].

### 7.2. IL-38 Impact on Immune Cell Differentiation and Function

IL-38 plays a crucial role in encouraging the differentiation of macrophages towards to a state that supports anti-inflammatory effects. A study by Ge at al. has shown that blocking the activity of IL-38 leads to increased inflammation and a shift in the spectrum of macrophage phenotypes toward those that suppress anit-inflammatory effects [106].

IL-38 also significantly impacts the crosstalk between macrophages and T cells. According to Mora et al., IL-38 inhibits the maturation and activation of Th17 cells, limiting the subsequent secretion of pro-inflammatory interleukin-17 (IL-17) [105]. Furthermore, IL-38 facilitates the proliferation of Tregs, which prevents their conversion into Th17 cells [103]. By restricting Th17 cell generation and IL-17 secretion, IL-38 helps mitigate excessive inflammation [105].

### 7.3. The Role of IL-38 in Ischemic Stroke

The treatment strategy utilizing tissue plasminogen activator (tPA), a thrombolytic agent, in patients with IS has been shown to significantly alleviate symptoms. In the study by Zare Rafie et al., a notable elevation in the concentration of IL-38 was observed 24 h after tPA administration compared to pre-treatment levels. Moreover, patients with less severe initial neurological deficits at admission experienced a greater increase in IL-38 levels after tPA. Additionally, a significant negative relationship was noted between changes in IL-38 levels, measured before and 24 h after tPA treatment, and the third-month functional outcome assessed based on the mRS score [104]. This inverse correlation between IL-38 levels and the severity of ischemic damage suggests that IL-38 may mitigate the deleterious effects of the cerebrovascular insult. The findings also highlight the potential role of IL-38 as a biomarker for predicting IS recovery and the response to thrombolytic therapy.

## 8. Limitations

The current understanding of the IL-1 family in the context of IS is limited by several factors. The immune responses mediated by IL-1 family cytokines can vary significantly depending on the specific combinations and contexts in which they are expressed. Additionally, the overlap in signaling pathways among IL-1 family members complicates the identification of their individual roles in inflammatory processes. Studying these cytokines in human brain tissue adds another layer of complexity due to the brain’s intricate structure and function. The difficulty in obtaining direct observations and samples from human brains further restricts our ability to fully comprehend the functions of IL-1 family members during IS. Moreover, the presence of comorbidities can alter interleukin levels, making it challenging to interpret their roles in the pathogenesis of IS accurately. Although growing evidence links IL-1 family cytokines to IS, substantial gaps remain in understanding their precise roles and potential therapeutic applications. Addressing these gaps is essential for developing targeted interventions that could improve outcomes for individuals affected by IS.

## 9. Future Perspectives

The role of IL-1 family cytokines in IS pathophysiology is a promising area of research that warrants further investigation. A deeper understanding of how these molecules influence IS, given their various subtypes, may enable more personalized management strategies. The exploration of IL-1-family-related biomarkers should focus on examining their effects at various time points following IS, both in the acute phase and long-term. These measurements need to be performed in the context of current standard treatments, such as tPA administration and MT. Additionally, exploring how genetic polymorphisms in IL-1 family-related genes influence IS susceptibility and outcomes could lead to more personalized prevention and treatment approaches.

The development and testing of IL-1 pathway inhibitors or modulators should consider various co-morbidities and patient characteristics to ensure that any potential therapies are safe and effective across a broad spectrum of patients with IS. An evaluation of the ability of these interventions to cross the BBB effectively is also necessary.

By pursuing these research directions, scientists and clinicians can advance our understanding of the IL-1 family’s role in IS pathophysiology. This knowledge could facilitate the development of more effective, targeted therapies that could significantly enhance outcomes for patients with IS.

## 10. Conclusions

The impact of IS on individuals and healthcare systems underscores the urgent need for groundbreaking early intervention strategies and refined prognostic tools. Currently, there is a notable absence of a reliable, user-friendly tools for assessing and accurately predicting IS outcomes. This deficiency represents a significant hurdle in our diagnostic and prognostic capabilities. To advance IS management, it is crucial to deepen our understanding of the complex mechanisms driving IS progression and recovery. These advancements are essential to enhance patient outcomes and alleviate the societal burden of IS.

A key area of focus lies in elucidating the delicate equilibrium between harmful and protective inflammatory processes. By exploring beyond conventional, well-established cytokines, we may uncover novel mechanisms that serve as promising biomarkers. The IL-1 family emerges as a particularly intriguing area of study. This diverse group of cytokines, encompassing both pro-inflammatory (IL-1a, IL-1b, IL-18 and IL-36) and anti-inflammatory (IL-1Ra, IL-33, IL-36A, IL-37 and IL-38) members, plays a crucial role in the post-IS inflammatory response (Figure 2). Their influence on both innate and adaptive immune responses, as well as IS risk, severity, and clinical outcomes, positions them as potential biomarkers for IS prognosis.

As our comprehension of the IL-1 family’s role in IS continues to expand, it opens up new possibilities for developing targeted diagnostic tools and therapeutic interventions. Moreover, the discovery that genetic variations in IL-1 genes may affect IS susceptibility emphasizes the importance of personalized treatment approaches.

The development of novel biomarkers and more tailored treatment strategies has the potential to significantly improve patient outcomes, which brings hope to millions affected by this condition.

## Figures and Tables

**Figure 1 ijms-25-10515-f001:**
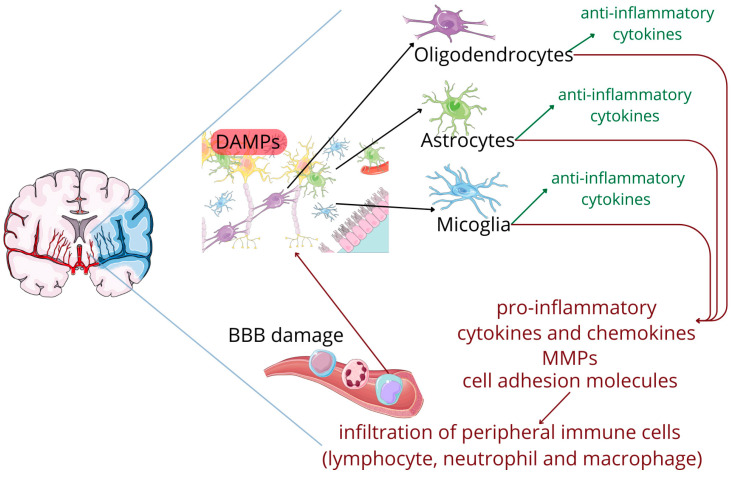
The cascade following an ischemic stroke. Figure abbreviations: BBB—blood brain barrier, DAMPs—damage-associated molecular patterns, MMPs—matrix metalloproteinases.

**Figure 2 ijms-25-10515-f002:**
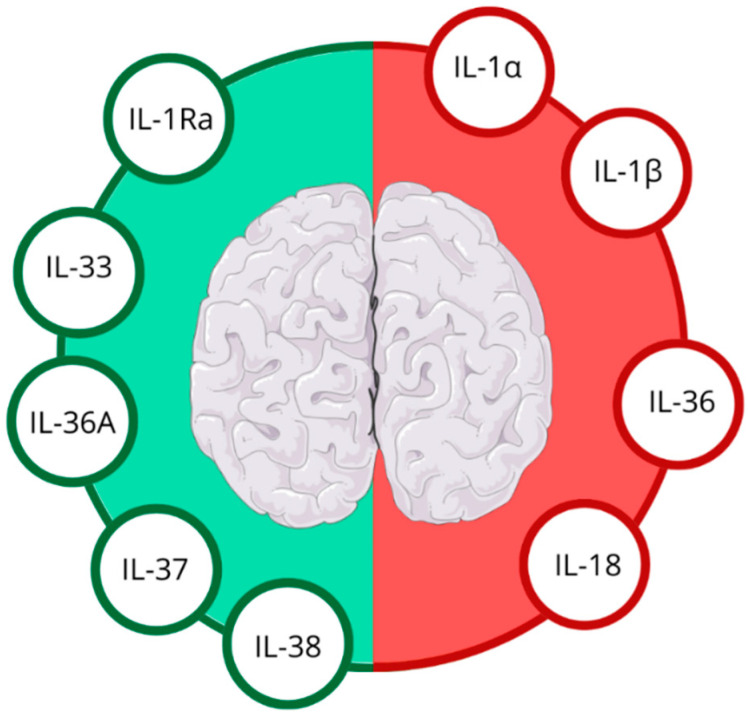
Anti-inflammatory (green half) and pro-inflammatory (red half) IL-1 family cytokines in ischemic stroke. Figure abbreviations: IL-1α—interleukin-1α, IL-1β—interleukin-1β, IL-1Ra—interleukin-1 receptor antagonist, IL-18—interleukin-18, IL-33—interleukin-33, IL-36—interleukin-36, IL-36Ra—interleukin-36 receptor antagonist, IL-37—interleukin-37, IL-38—interleukin-38.

**Figure 3 ijms-25-10515-f003:**
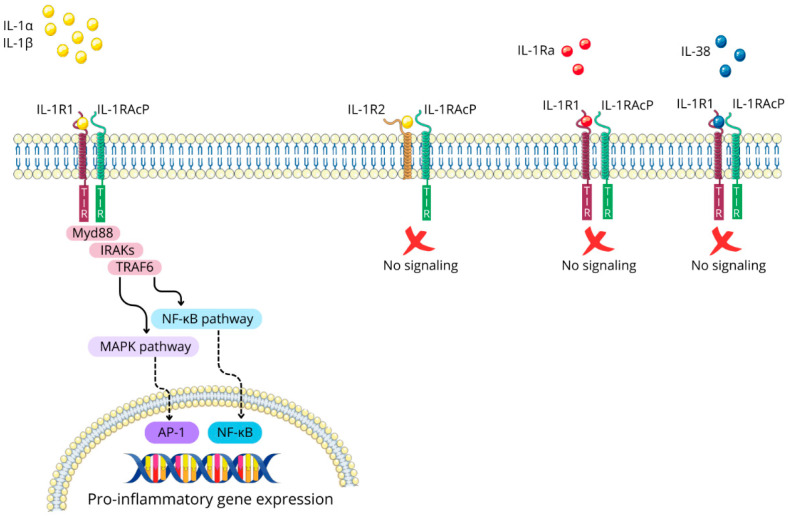
The IL-1 signaling pathway and its regulatory mechanisms. Figure abbreviations: AP-1—activator protein 1; IL-1α—interleukin-1α; IL-1β—interleukin-1β; IL-1R1—interleukin-1 receptor type 1; IL-1Ra—interleukin 1 receptor antagonist; IL-1RAcP—interleukin-1 receptor accessory protein; IL-1R2—interleukin-1 receptor type 2; IL-38—interleukin-38; IRAK—interleukin-1 receptor-associated kinases; MAPK—mitogen-activated protein kinases; Myd88—myeloid differentiation primary response protein 88; NF-κB—nuclear factor-kappa B; TIR domain—the toll-interleukin-1 receptor homology domain; TRAF6—tumor necrosis factor receptor associated factor 6.

**Figure 4 ijms-25-10515-f004:**
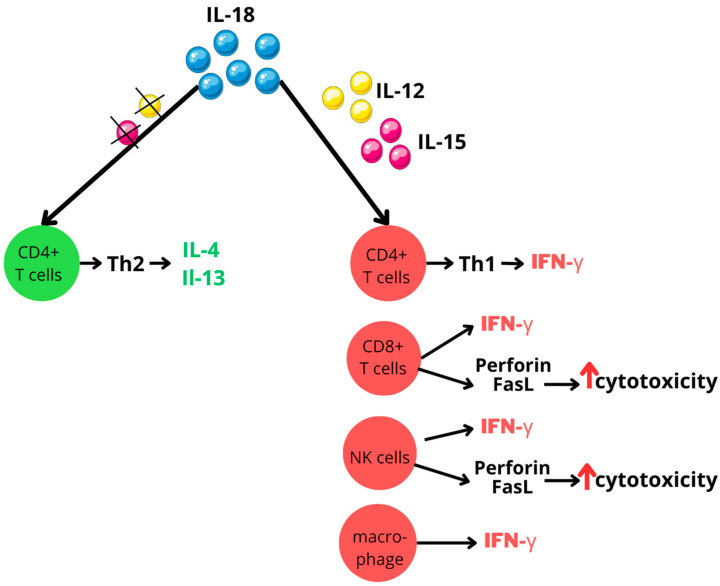
IL-18 activity modulated by IL-12 and IL-15 interplay. Figure abbreviations: CD—cluster of differentiation; FasL—Fas ligand; IFN-γ—interferon-gamma; IL—interleukin; NK—natural killer; Th—T helper; ↑—increased.

**Figure 5 ijms-25-10515-f005:**
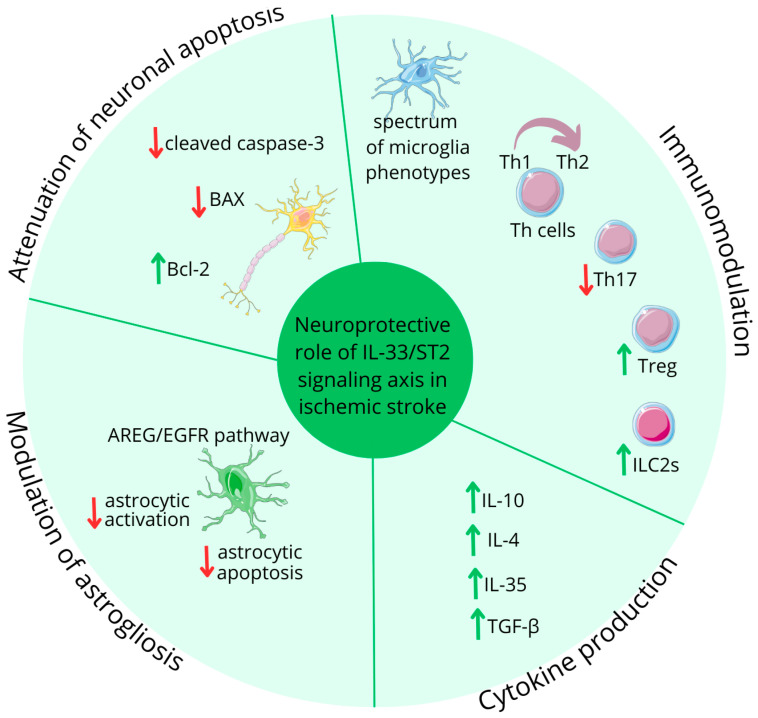
IL-33 role in ischemic stroke. Figure abbreviations: AREG—amphiregulin; EGFR—epidermal growth factor receptors; IL—interleukin; ILC2s—group 2 innate lymphoid cells; TGF-β—transforming growth factor β; Th—lymphocyte T helper; Treg—regulatory T cells; ↓—decreased; ↑—increased.

**Figure 6 ijms-25-10515-f006:**
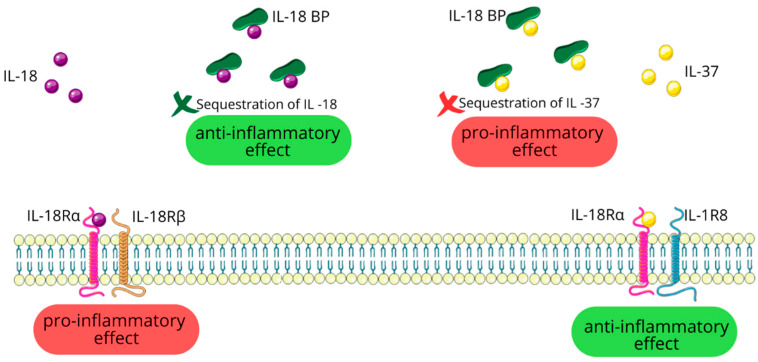
IL-37 and IL-18: shared receptors, opposing functions. Figure abbreviations: IL-18—interleukin-18; IL-18BP—IL-18 binding protein; IL-18Rα—IL-18 receptor alpha chain; IL-18Rβ—IL-18 receptor beta chain; IL-1R8—IL-1 receptor 8; IL-37—interleukin-37.

**Table 1 ijms-25-10515-t001:** IL-family cytokines, its receptors, and cellular sources.

Cytokine	Receptor	Co-Receptor	Cellular Sources
IL-1α	IL-1R1	IL-1RAcP (IL-1R3)	Macrophages, monocytes, platelets, microglia, astrocytes, endothelial cells, keratinocytes, fibroblasts, intestinal epithelial cells, other cells comprising kidney, liver, lung [12,13,14]
IL-1β	IL-1R1	IL-1RAcP (IL-1R3)	Macrophages, monocytes, leukocytes, dendritic cells [12,13,14]
IL-1Ra	IL-1R1		Macrophages, neutrophils, epithelial cells, fibroblasts, keratinocytes, hepatocytes [15]
IL-18	IL-18Rα (IL-1R5)	IL-18Rβ (IL-18RAP)	Macrophages, monocytes, microglia, dendritic cells, endothelial cells, keratinocytes, intestinal epithelial cells, neurons, osteoblasts, mesenchymal cells [16,17,18]
IL-33	ST2L (IL-1RL1, IL-1R4)	IL-1RAcP (IL-1R3)	Oligodendrocytes, astrocytes, keratinocytes, epithelial cells, endothelial cells from blood vessels, lymphatic endothelium, fibroblastic stromal cells [19,20]
IL-36α, β, γ	IL-1Rrp2 (IL-1R6)	IL-1RAcP (IL-1R3)	Macrophages, monocytes, B cells, T cells, dendritic cells, bone marrow cells, keratinocytes, neurons, glial cells, intestinal epithelial cells, bronchial and lung epithelial cells, synovial cells, other cells comprising secondary lymphoid organs, heart and testis [21]
IL-36Ra	IL-1Rrp2 (IL-1R6)		Epithelial cells, keratinocytes, dendritic cells, B cells, macrophages, monocytes [21,22]
IL-37	IL-18Rα (IL-1R5)	IL-1R8 (SIGIRR, TIR8)	Monocytes, macrophages, dendritic cells, tonsil B cells, T cells and plasma cells, bone marrow cells, endothelial cells, keratinocytes, intestinal epithelial cells, lung epithelial cells, other cells comprising heart, kidney, lymph node, thymus, testis, placenta, and uterus [23,24]
IL-38	IL-1Rrp2(IL-1R6); IL-1R1; IL1RAPL1		Macrophages, monocytes, B cell, cytotoxic T cell, apoptotic cells, epithelial cells, endothelial cell, fibroblast, keratinocytes, parictal cells, chief cells, Langerhans cells, other cells comprising secondary lymphoid organs, thymus, heart, and placenta [25,26]

Table abbreviations: IL-1α—interleukin-1α; IL-1β—interleukin-1β; IL-1R1—interleukin-1 receptor type 1; IL-1R3—interleukin-1 receptor type 3; IL-1R4—interleukin-1 receptor type 4; IL-1R5—interleukin-1 receptor type 5; IL-1R6—interleukin-1 receptor type 6; IL-1R8—interleukin-1 receptor type 8; IL-1Ra—interleukin 1 receptor antagonist; IL-1RAcP—interleukin-1 receptor accessory protein; IL-1RAPL1—interleukin-1 receptor accessory protein-like 1; IL-1RL1—interleukin 1 receptor like 1; IL-1Rrp2—IL-1 receptor-related protein 2; IL-18—interleukin-18; IL-18Rα—IL-18 receptor alpha chain; IL-18Rβ—IL-18 receptor beta chain; IL-18RAP—interleukin 18 receptor accessory protein; IL-33—interleukin-33; IL-36—interleukin-36; IL-36Ra—interleukin-36 receptor antagonist; IL-37—interleukin-37; IL-38- interleukin-38; SIGIRR—single immunoglobulin IL-1R-related receptor; ST2L—membrane-bound form of ST2; TIR8—Toll IL-1 receptor 8.

## Data Availability

Not applicable.

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
