# Peer review of "Expanding Role of Interleukin-1 Family Cytokines in Acute Ischemic Stroke"

_ijms, 2024, doi:10.3390/ijms251910515_

Round 1

Reviewer 1 Report

Comments and Suggestions for Authors

The review article by Paulina Matys and colleagues thoroughly reviews the expanding role of Interleukin-1 family cytokines in Acute Ischemic Stroke (IS). This review provides an updated summary of the IL-1 family’s involvement in IS, consisting of pro-inflammatory molecules (IL-1α, IL-1β, IL-18, and IL-36) and anti-inflammatory molecules (IL-1Ra, IL-33, IL-36A, 16 IL-37, and IL-38). This evolving role of IL-1 has opened new approaches for diagnostic and therapeutic strategies, and this review highlights IL-1 role in IS modulation and identifies promising IL-1-related biomarkers. Overall, this review article is well-written and significantly emphasizes the progressive role of IL-1 family cytokines in IS and the potential of these IL-1 cytokines in the treatment. The article further explores the mechanism of action for each IL-1 discussed here and provides a critical link to various aspects of IS. The overall quality of the manuscript could be improved, and the following changes are needed in the article.

In the introduction section, the authors could discuss more on the role of IL-1 in various diseases highlighting the IL-1 molecular role.  

The font sizes in figure1 should be increased for better readability. 

Minor corrections are required for the sentence on the line 530-531. 

It would be a good idea to include a table for the tissue distributions of the IL-1 family and the molecules described in this review (pro-inflammatory and anti-inflammatory). 

In the future direction section, the authors have highlighted the importance of the role of IL-1 family cytokines in IS pathophysiology and the possibility of developing more effective, targeted therapies for improved patient outcomes. The review article must mention the limitations of the current understanding of the IL-1 families in IS.

Reviewer 2 Report

Comments and Suggestions for Authors

The review by Paulina Matys and colleagues is a well written and extensive summary of current knowledge on the role of IL-1 family cytokines in acute ischemic stroke. It captures the current understanding well. The figures are well designed and help with the comprehension of the text. Some minor issues are listed below.

- I might have missed it in the references provided, but I cannot see where it says that stroke “ranks as the second leading cause of death worldwide”. Can the authors please point me to the corresponding section in references 3 and 4?

- Figure 1: the concept of M1 versus M2 microglia is outdated. Rather there is a spectrum of phenotypes. I urge the authors to remove “M1” and “M2.  Also, some of the labels in Figure 1 are small.

- line 40-41: isn’t it also conceivable that a stroke directly damages BBB integrity?

- line 159: Liberale et al. is mentioned as reference but not actually references until line 169. It  would be good to include the reference at time of first mentioning.

- line 170: as for line 159. Please check also other instances throughout the manuscript.

- line 202: “exhibit increased higher …levels”. Consider removing increased or higher.

- line 204: “-511genes” please correct

 - line 239-240: does this include microglia?

- line 258-260 and Figure 4: rather than saying that Il-4 or Th2 is anti-inflammatory, I suggest considering the use of “immunomodulatory”.

- figure 5 and text: revisit M1 vs. M2 microglia
